# Development and Evaluation of 2-Amino-7-Fluorophenazine 5,10-Dioxide Polymeric Micelles as Antitumoral Agents for 4T1 Breast Cancer

**DOI:** 10.3390/polym14010071

**Published:** 2021-12-25

**Authors:** Nicole Lecot, Belén Dávila, Carina Sánchez, Marcelo Fernández, Mercedes González, Pablo Cabral, Hugo Cerecetto, Romina Glisoni

**Affiliations:** 1Laboratorio de Radiofarmacia, Centro de Investigaciones Nucleares, Facultad de Ciencias, Universidad de la República, Mataojo 2055, Montevideo 11400, Uruguay; pabloc7@gmail.com (P.C.); hcerecetto@cin.edu.uy (H.C.); 2Grupo de Química Orgánica Medicinal, Instituto de Química Biológica, Facultad de Ciencias, Universidad de la República, Mataojo 2055, Montevideo 11400, Uruguay; BelenDav@gmail.com (B.D.); caarisanchez@hotmail.com (C.S.); mgonzalezhormaizteguy@gmail.com (M.G.); 3Laboratorio de Experimentación Animal, Centro de Investigaciones Nucleares, Facultad de Ciencias, Universidad de la República, Mataojo 2055, Montevideo 11400, Uruguay; xxmferna@gmail.com; 4Departamento de Tecnología Farmacéutica, Facultad de Farmacia y Bioquímica, Universidad de Buenos Aires, Junín 956, Buenos Aires C1113AAD, Argentina; 5Instituto de Nanobiotecnología (NANOBIOTEC), CONICET-Universidad de Buenos Aires, Junín 956, Buenos Aires C1113AAD, Argentina

**Keywords:** bioreductive-drug, phenazine-5,10-dioxide, amphiphilic pristine polymeric micelles, 4T1-tumor model, in vivo antitumoral activity

## Abstract

2-Amino-7-fluorophenazine 5,10-dioxide (FNZ) is a bioreducible prodrug, poorly soluble in water, with potential anticancer activity on hypoxic-tumors. This poor solubility limits its potential applications in clinic. Amphiphilic pristine polymeric micelles (PMs) based on triblock copolymers Pluronic^®^ and Tetronic^®^, glycosylated derivatives and their mixtures with preformed-liposomes (LPS), were analyzed as strategies to improve the bioavailability of FNZ. FNZ encapsulations were performed and the obtaining nanostructures were characterized using UV-visible spectroscopy (UV-VIS), Transmission Electron Microscopy (TEM) and Dynamic Light Scattering (DLS). The most promising nanoformulations were analyzed for their potential toxicity and pharmacologically, at 20 mg/kg FNZ-doses, in a stage-IV murine metastatic-breast tumor model. The results revealed that the solubility of the encapsulated-FNZ increased up to 14 times and the analysis (UV-VIS, DLS and TEM) confirmed the interaction between vehicles and FNZ. In all the cases appropriate encapsulation efficiencies (greater than 75%), monodisperse nanometric particle sizes (PDI = 0.180–0.335), adequate Z-potentials (−1.59 to −26.4 mV), stabilities and spherical morphologies were obtained. The in vitro profile of FNZ controlled releases corresponded mainly to a kinetic Higuchi model. The in vitro/in vivo biological studies revealed non-toxicity and relevant tumor-weight diminution (up to 61%).

## 1. Introduction

Breast cancer is the second leading cause of mortality in women worldwide and it has been increasing in recent years [1]. Particularly, estrogenic receptor negative tumors (ER-) have a more aggressive, malignant and metastatic phenotype, related to a worse prognosis. Especially, triple negative breast tumors (TNBC), such as 4T1 mammary carcinoma mouse model, are the most aggressive and those associated with higher mortality. The treatment still constitutes a great challenge, mainly involves radiotherapy, chemotherapy and surgery. Furthermore, altered tumor metabolism and enzymes regulated by hypoxia factor 1a (HIF-1a) are vital for the process of tumor progression, angiogenesis and metastasis [1,2,3].

Our research group has developed and described phenazine 5,10-dioxide derivatives, as selective agents in hypoxic conditions, from which leaders emerged to treat solid tumors [4,5,6,7,8,9,10,11,12]. Particularly, the overexpression of reductase enzymes participates in biological processes by bioreducting compounds with these *N*-oxide functions. When the *N*-oxide is bioreduced via electron process, a cytotoxic metabolite is generated, the nitroxide radical. If the metabolization is by reduction of the *N-*oxide to amines, via two electrons, it leads to an agent that could interact with biomolecules like DNA, RNA polymerases and topoisomerases or both and the amine-protonation promotes an exclusively confined in the diseased tissue due to the acidic pH of the tumor cell microenvironment in the hypoxic conditions [13,14]. Derivatives of phenazine dioxides, such as the compound 2-amino-7-fluorophenazine 5,10-dioxide (FNZ), have low aqueous solubility (208 µg/mL) that prevents their selective action at the tumor level, their oral administration, parenteral and transdermal [11,15,16]. Thus, its clinical use would be restricted, due to the low biodistribution of the body aqueous milieu.

Fortunately, nanomedicine has emerged as a tool for solving solubility problems [16]. This discipline involved the development of systems, known as nanosystems, carriers or vehicles, that additionally allow passive transport to the tumor microenvironment, due to inconsistent blood vessels (diameter, irregular shape, abnormal protrusions and blind ends), absence of supporting tissues of the vasculature, vascular hyperpermeability, fenestrations of 100 nm to 2 µm in diameter, poor lymphatic system and a thermodynamically favorable environment in terms of retention of nanoparticles, as long as they are smaller than endothelial “hollow” pores. Therefore, all of these characteristics generate an effect at the level of the tumor microenvironment known as the Enhanced Permeability and Retention (EPR) effect [17,18,19,20] and it is presented as one of the strategies for the delivery of therapeutics to different types of cancer. In the past few years, the molecular biology of breast cancer allowed the development of strategies that could be use novel nanotherapeutics. In order to do this, the nanosystems are promising alternative particularly for the treatment of breast cancer [21,22,23].

The polymeric micelles (PMs) are biocompatible material that were first proposed as nanovehicles for drug delivery by Ringsdorf in 1984 has since been widely developed according their performance [24,25]. Amphiphilic PMs are spherical with core/shell architecture formed by the spontaneous self-assembly in water above a critical micellar concentration (CMC) [26,27]. The core consists of a hydrophobic segment and the hydrophilic segment conforms the shell or corona allowing incorporating complete or partially hydrophobic molecules with biological activity [28,29,30,31,32].

Herein in the present work, we studied the capability of self-assembled amphiphilic PMs and mixed PMs based of pristine block copolymers of polyethyleneoxide-polypropyleneoxide (PEO-PPO) and their glycosylated derivatives [33,34,35,36,37], to improve the hydrosolubility and bioavailability of FNZ as possible candidate agent for the treatment of breast cancer. The glycosylation of PMs was previously for us, to promote the potential micelles-accumulation within the tumor as the result of a greater cellular uptake at the expense of the exacerbated increase in glucose metabolism in tumor cells and overexpression of GLUTs receptor [34,35,36,37,38,39].

## 2. Materials and Methods

### 2.1. Materials

Pluronic^®^ 127 (127; molecular weight: 12,600 g/mol; PEO content 70% by weight) and Tetronic^®^ 1307 (T1307; molecular weight: 18,000 g/mol; PEO content 70% by weight) were supplied by BASF Corporation (New Mildorf, CT, USA) and used as received. Unilamellar preformed liposomes (LPS) composed by cholesterol (2 mg/mL), hydrogenated soy phosphatidylcholine (6 mg/mL) and phosphatidylethanolamine (4 mg/mL) grafted to a short chain of polyethylene glycol (*M_w_*~2 kDa) were gently gifted by MR-Pharma S.A. (Buenos Aires, Argentina). Dimethylsulfoxide (DMSO, analytical quality) was purchased from Sigma-Aldrich (St. Louis, MO, USA). RPMI milieu was purchased from Capricorn Scientific (Ebsdorfergrund, Germany). In all the cases, the water was purified and deionized (18 MΩ/cm^2^) in a Milli-Q water filtration system (Millipore Corp., Milford, CT, USA).

### 2.2. Animals

BALB/c female mice weighing 18–20 g were produced and provided by Unidad de Reactivos para Biomodelos de Experimentación (URBE), Facultad de Medicina, Universidad de la República, Uruguay. The authors state that they followed the principles outlined in the Declaration of Helsinki for all animal experimental investigations. Animals were housed in wire mesh cages at 20 ± 2 °C with 12 h artificial light-dark cycles. The animals were fed ad libitum to standard pellet diet and water and were used after a minimum of 3 days of acclimation to the housing conditions.

All protocols for animal experimentation were carried out in accordance with procedures authorized by the Ethical Committee for Animal Experimentation, Uruguay, by whom this project was previously approved (CHEA-UdelaR Protocol number 240011-002249-15).

### 2.3. Methods

#### 2.3.1. UV/Vis Spectrophotometry for Determination of Intrinsic Solubility of FNZ in Water and Apparent Solubility of FNZ in Presence of PMs

Lineal calibration curve of FNZ [9], in the range of 2.5–12 μg/mL, was obtained in DMSO measuring at 296 nm (UV/Vis spectrometer model Evolution 160 UV-Vis, Thermo-Scientific, Waltham, MA, USA) (Appendix A). The curve was determined in triplicate (three independent samples, R^2^ = 0.9911).

#### 2.3.2. Preparation of PMs and PMs:LPS Loaded and Unloaded with FNZ

The preparation of pristine and glucosylated (Glu) poly(ethyleneoxide)-*b*-poly(propyleneoxide) (PEO-PPO) PMs in aqueous or lipid dispersions (LPS), loaded or unloaded with FNZ, was carried out as we previously reported [32,33,34,35]. Briefly, 750 mg of pristine F127, F127-Glu [34,35], T1307-Glu [35], or T1307:F127 (50:50) were hydrated with 2.5 mL of H_2_O at room temperature. The preparation was left overnight at 4 °C and then the volume was completed, at room temperature, to 5.0 mL (final copolymer concentration of 15% *w*/*v*). The same procedure was carried out in the presence of preformed LPS [40] to increase the solubilizing capacity of FNZ and to be able to study later the thermosensitivity behavior of these smart materials in the presence of LPS. For this, LPS dispersion (2.5 mL) was used to hydrate 300 mg of the copolymer F127 at 4 °C overnight. Then, the volume was completed, at room temperature, to a final volume of 3.0 mL (final copolymer concentration of 10% *w*/*v*).

Finally, in both cases, FNZ was added in excess (up 3 mg/mL as appropriate), the preparations were stirred for 24 h at 200 rpm, on a multipoint shaker, at room temperature and finally, clarified using clarification cellulose acetate membrane filters to remove the excess of non-encapsulated FNZ. The nanosystems obtained were called F127-Glu/FNZ, T1307-Glu/FNZ, T1307:F127/FNZ and F127:LPS/FNZ, respectively.

#### 2.3.3. Characterization by Dynamic Light Scattering (DLS)

DLS measurements were made at 25 °C using a Zetasizer Nano-ZS equipment (Malvern Instruments, Malvern, UK), equipped with a He-Ne laser (633 nm) and a digital correlator, model ZEN3600. The measurements were made at a dispersion angle of 173° and fixed laser light of 4.65 mm. Three independent experiments were performed. Data were expressed as mean ± standard deviation using the Nano-ZS software (v7.12 software, Malvern Instruments). Particle size (D*_h_*), Zeta potential (Z-pot) and polydispersity index (PDI) were determined.

#### 2.3.4. Morphological Analysis by Transmission Electron Microscopy (TEM)

Free-PMs (40 µL) and FNZ-loaded PMs (40 µL) were deposited on an ultrathin carbon film on QUANTIFOIL holey support film copper grid (200 mesh, Electron Microscopy Sciences, Hatfield, PA, USA) in a bottom Elisa plate flat and incubated 1 h at 37 °C. The samples were covered with phosphotungstic acid (2%) for 60 s. Finally, grids were dried at room temperature for 8–10 min. The samples images were taken in two screens of TEM (Jeol, JEM-1010 transmission electron microscope, JEOL, Tokyo, Japan) and were processed with ImageJ (National Institutes of Health, Bethesda, MD, USA) and Gimp 8.0 software. Magnification: 6000×, except (G) 185,000×.

#### 2.3.5. In Vitro Release of FNZ from the Nanoformulations

FNZ loaded PMs (5.0 mL) were placed into dialysis membranes (MWCO of 3500 g/mol, Spectra/Por^®^, Spectrum Laboratories, Inc., Rancho Dominguez, CA, USA) with 40.0 mL of H_2_O and were left in constant stirring at 25 °C. After 24 h a total volume of 200.0 mL was completed with H_2_O. The release of FNZ was analyzed at 10, 20, 30, 45, 90 and 120 and 1440 min. The samples (50–150 μL) were taken outside the membrane, were diluted to 1.0 mL with DMSO and analyzed by UV-Vis at 296 nm. The tests were carried out in triplicate and the results were expressed as a % of cumulative FNZ released as function of time [41,42]. Finally, zero order, first order, Higuchi and Korsmeyer–Peppas kinetic models were examined. The correlation coefficients (R^2^) were calculated in each case for obtaining the best fitted model to explain the release kinetics and mechanism of FNZ release.

### 2.4. Preclinical Toxicity Studies

#### 2.4.1. AMES Test

The direct mutagenicity of nanoformulations in *Salmonella* typhimurium auxotrophs for the histidine operon, strains TA98, TA100, TA1535, TA1537 and TA102, were determined, in triplicate, using the maximum tolerated doses and four consecutive dilutions to the third according to the Organization for Economic Cooperation and Development (OECD) recommendations [43]. The average of the number of mutated colonies was graphed in function of the concentration and the mutagenic powers (MPs, slope of the initial linear portion) were extracted for each tested strain [44,45,46,47].

#### 2.4.2. Intraperitoneal (IP) Acute Toxicity Studies

The intraperitoneal LD_50_ determination was carried out by “Up and Down” test according to OECD recommendations [48,49]. Mice were intraperitoneal administered with FNZ loaded PMs, in adequate doses (185 µL), using free FNZ (in DMSO, 4%) as comparison. In particular, the “limit test” was used in which a maximum of 5 animals were used sequentially starting with a dosage of 2000 mg/kg and analyzing during 72 h the vital signs (Irwin Test) [50,51] and determining LD_50_ with the software AOT425StatPgm [52,53]. Subsequently, the animals were anesthetized with isoflurane and sacrificed by cervical dislocation performing the necropsies of the animals in order to see macroscopic physiological changes (adhesions and/or macroscopic changes of the organs).

#### 2.4.3. In Vivo Antitumor Efficacy

The 4T1 mammary carcinoma cell line [54], in RPMI milieu, was inoculated (700,000 cells/100 µL) subcutaneously in the fourth mammary gland of the mice, taking as a reference the nipple of the left pre-inguinal region. After 5 days of inoculation, tumor diameters were measured daily with a sterile vernier caliper. The two tumor diameters, width and length, were measured and the tumor volume (V) was calculated using the following equation: V = (width^2^ × length)/2 [15,54,55,56,57].

The treatments with FNZ-loaded PMs began on day 5 after inoculation of the cells (since the animals present measurable, with calipers, tumors).

The following treatment groups were defined: (i) F127-Glu, T1307-Glu, T1307:F127, or F127:LPS PMs with FNZ (at 20 mg/kg of body weight per IP injection) (animals per treatment, *n* = 6); (ii) Negative control (IP injection of H_2_O) (animals per treatment, *n* = 6). Days of treatments (after 4T1-inoculations): 5th, 7th, 9th, 12th, 14th and 16th (Figure 1). Tumor volumes were determined daily. At day 18, each animal was subjected to gross necropsy. All gross pathological changes were recorded for each animal. Blood for biochemical and hematological studies was drawn by sectioning the subclavian artery maintaining the blood in EDTA or heparin at 0 °C. For biochemical and hematological studies healthy and untreated animals were included.

The changes in tumor size and the animals’ survival percentages were recorded during the study. For the survival studies Kaplan–Meier analysis were carried out with the non-parametric log-rank test with a *p* < 0.05 (0.036) and X^2^ > 3.84 (6.818). The efficacies of the treatments were also evaluated by the weight of the tumors at the end of the treatments.

#### 2.4.4. Statistical Analysis

Statistical analysis was performed using Origin 8.0 program and Excel 2010 (Student’s *t* test in a comparison between two groups) and Graphpad 6.0 (GraphPad Software Inc., San Diego, CA, USA) with the ANOVA “two way” test followed by Tukey’s test. The differences between the evaluated parameters were statistically significant if the *p* value is less than 0.05 and very significant if the *p* value is less than 0.01.

## 3. Results

### 3.1. Nanoformulations: Preparations and Characterizations

The following nanoformulations were successfully prepared: FNZ-loaded within F127-Glu, T1307-Glu, T1307:F127, or F127:LPS (F127-Glu/FNZ, T1307-Glu/FNZ, T1307:F127/FNZ, or F127:LPS/FNZ). For all PMs an increment of FNZ solubility was observed (Table 1) up to 14 times respect the intrinsic solubility of FNZ in Milli-Q water (208 μg/mL, Table 1). Noticeably, the profile of the solubility was related with the level of FNZ-encapsulations which was high for all the formulations being T1307-Glu PMs, the polymeric nanosystem with the lowest encapsulation-capability (~74%, Table 1), while the other nanosystems displayed %EE between 87 and 95% (Table 1).

Free-PMs and PMs with FNZ cargo showed hydrodynamic sizes between 60 and 700 nm (Table 1, major populations %, Appendix A) and in general sizes of PMs decreased after the encapsulation of FNZ. Instead, F127-Glu/FNZ PMs displayed an increase in size (~700 nm) with a unique monomodal population, after encapsulation of FNZ (Table 1). On the other hand, PDI is a parameter that indicates the degree of polydispersity in the sample [32,33,34,35]. All PDI values obtained, corresponded to values less than 0.5, between 0.180 and 0.335 (Table 1), indicating that PMs/FNZ were all suitable for monomodal distribution.

Z-pot is a relevant parameter of the degree of stability of the samples [32,33,34,35]. The Z-pot values for F127-Glu, T1307-Glu and T1307:F127 varied between −1.6 to −13.9 mV (Table 1), which it is compatible with biological membranes for subsequent endocytosis. In addition, the mixed F127:LPS (Table 1) resulted in more negative Z-pot nanosystems, in the ranged of −16 and −26 mV, which is very promising for use in biological milieu, indicating that LPS stabilized the micellar corona conferring greater physicochemical stability and being able to decrease the tendency of PMs self-aggregation [34,37].

Additionally, TEM studies were performed in order to visualize the nanosystems morphologies. The microscopies showed nanometric spherical structures in one dimension (Figure 2 and Appendix A) with an average size in agreement to DLS experiments.

### 3.2. FNZ Release from the Nanoformulations

In all the cases a controlled and prolonged FNZ release, from the nanostructures, during the slow release-phase was confirmed probably as result of the nanosystems greater surface area [41,42,58,59]. In general, the release in the first hour increased from 2.5 to 8% indicating adequate FNZ-nanosystems interactions, FNZ-complex inclusion, then a slow release was observed for at least 24 h. After 48 h, 100% FNZ-release was observed in all the cases. Except for the formulation with F127-Glu, the best correlations corresponded to a Higuchi model (Table 2), with a release mechanism controlled primarily by diffusion of FNZ from de PMs. For F127-Glu/FNZ formulation, the best fit was obtained with a First, Order prolonged release (Table 2). According to the Korsmeyer–Peppas model and “*n*” coefficient, the Fickian diffusion type (Type I) displays values of *n* < 0.43 and the drug release is governed only by the drug diffusion. With “*n”* values between 0.43 and 0.85 (as T1307:F127/FNZ and F127:LPS/FNZ PMs), non-Fickian anomalous type release, the drug release can be attributed to a combination of mechanisms: (i) the diffusion of the drug and (ii) the relaxation of the copolymer chains as the medium diffuses through the polymer matrix and for values of *n* ≥ 0.85 (as for glucosylated PMs: F127-Glu/FNZ and T1307-Glu/FNZ, Table 2), FNZ release is purely governed by the copolymeric relaxation (Type Supra II Non-Fickian model) [41,42].

### 3.3. Preclinical Toxicity Results

In order to further draw information about the potentiality of formulations as therapeutic tools, supplementary studies related to toxicity were conducted. We centered our attention on some of the OECD recommendations, in special information about in vitro mutagenicity and in vivo-LD_50_, were explored [43,48,49].

For the in vitro mutagenicity the AMES test was performed for free FNZ and FNZ loaded PMs using *Salmonella* typhimurium TA98, TA100, TA1535, TA1537 and TA102 strains. The samples were considered mutagenic when the number of reverted colonies was at least twice of the negative control for at least two consecutive dose levels [44] and the MPs were determined. The MP of FNZ was modified after encapsulation while free FNZ displayed the highest MP (Figure 3), evidenced mainly in TA98 and TA100 strains, the nanoformulations have always been less mutagenic. Especially, F127-Glu/FNZ was not mutagenic against TA100 strain possibly due to the protection of the FNZ by the polymeric micelles or by the prolonged FNZ-release from the formulation (Figure 3).

On the other hand, acute toxicity was performed by IP administration using modified protocol of OECD [48,49,53]. This study showed that encapsulated FNZ in all the formulations had LD_50_ greater than 2000 mg/kg of mouse body weight (Table 3). This showed a significant improvement in safety respect to free FNZ (LD_50_ = 1500 mg/kg of mouse body weight). Additionally, the macroscopic analysis after the necropsy showed that there were no organs adhesions or apparent macroscopic physiological changes after the posology.

### 3.4. In Vivo Antitumor Results

In vivo treatments of BALB/c mice-bearing 4T1-breast tumor evidenced antitumor activity in all nanoformulations particularly with the glycosylated formulations (T1307-Glu/FNZ and F127-Glu/FNZ) with a drop in tumor sizes, respect to the control, at the end of the treatments (Figure 4). Additionally, the treatment with FNZ vehiculizated with T1307:F127 nanoformulation showed the lowest tumor volume during the assay (Figure 4).

When at the end of the assays the mice necropsies were performed a particular behavior of tumor tissues was observed, i.e., shelled tumor, non-solid, for animals treated with the FNZ loaded nanoformulation F127:LPS (Figure 5). Consequently, in addition to the assessment of changes in the tumor size, we evaluated the tumor weights in order to know the tumor fluffiness. In this analysis, we again evidenced that FNZ loaded into glycosylated polymeric micelles (F127-Glu/FNZ and T1307-Glu/FNZ) provoked the best antitumor performances having the other assayed nanoformulations/FNZ, i.e., T1307:F127/FNZ and F127:LPS/FNZ, remarkable behaviors (Figure 5b). The tumor masses for F127-Glu/FNZ and for T1307-Glu/FNZ treatments were near to 37% of the control-average tumor masses.

Additionally, the animal survival was analyzed by the Kaplan-Meier method [60]. Treatments with T1307-Glu/FNZ and T1307:F127/FNZ showed 100% animal survival during the all period of the assay (Figure 6). The other two nanoformulations, i.e., F127-Glu/FNZ and F127:LPS/FNZ, reached about 60% at the end of the treatments.

Furthermore, the biochemical and hematological findings (Table 4) were in agreement to the necropsies, i.e., absence of organ damages. In the treated animals, the studied parameters were restored, comparing to untreated animals and were in agreement with the evolution of tumor sizes, showing a potential improvement in the animals with chemotherapies. Treatment with T1307-Glu/FNZ showed normal levels in all the evaluated parameters, except for urea in blood. In all the treatments, alanine aminotransferase (GPT) and aspartate aminotransferase (GOT) values were lower than the corresponding cut-offs (normal value for GPT = 40–170 UI/L; normal values for GOT = 67–381 UI/L [61]) indicating reestablishment of normal hepatic functions, comparing to untreated animals that could have altered liver-metabolism due to the metastatic breast cancer.

## 4. Discussion

The development of nanosystems to transport drugs to invasive tumors is one of the areas of greatest development, with the EPR effect being the most described passive mechanism for the accumulation of nanomaterials in tumors (liposomes, PMs, dendrimers, quantum dots, among others). In this sense, herein we studied nanotechnological strategies to overcome solubility obstacle of poorly soluble drug FNZ and taking advantage of EPR effect to increase its bioavailability in hypoxic tumors. PMs based triblock copolymers of PEO-PPO, nanostructures that present a core/corona architecture composed of amphiphilic blocks, were useful to FNZ transport and delivery. The FNZ loaded PMs were obtained in a controlled, reproducible and scalable via, presenting a nanostructured-sphere structure with adequate hydrodynamic diameters, suitable sizes and Zeta potentials, evidenced by DLS and TEM. The encapsulations, greater than 75%, overcame FNZ poor aqueous solubility. In addition, we evaluated the capability and the beneficial effect of polymeric micelle glycosylation for the active targeting in glucose transporter-expressing adult tumors, as well as their mixtures in aqueous and lipid dispersions.

The FNZ releases from the nanostructures, following in general a Higuchi model, were controlled and prolonged during the phase of slow release. This slow release may be due to the inclusion of the FNZ complex in PMs that confirms an optimal interaction between FNZ and PMs. This slow release could generate an increase of FNZ concentration in plasma circulation and gives a greater probability of tumor uptake. Therefore, this increment in circulation could impact on the reduction of dosage frequencies.

Additionally, the preclinical toxicity studies showed, on the one hand, that all the nanoformulations displayed lower mutagenic powers, in the Ames test, than free FNZ. Especially, the FNZ-loaded into F127:LPS, F127-Glu or T1307-Glu showed significant diminution of mutagenic power respect to free FNZ resulting the F127-Glu/FNZ system not mutagenic against one of the studied strain. On the other hand, the acute intraperitoneal toxicities of the nanoformulations, expressed as LD_50_, were lower than the free FNZ.

Finally, the in vivo treatments of BALB/c mice bearing 4T1 breast tumor evidenced antitumor activity in all the nanoformulations. The best performance nanoformulation was T1307-Glu/FNZ which displayed the lowest tumor volume and mass at the end of the assay, the highest percentage of animals’ survival and restoring of abnormal biochemical and hematological findings. The good antitumoral activity of this FNZ loaded glycosylated nanosystem could be result of the: (1) EPR effect that depends on the permeable nature of the tumor vasculature and prolonged circulation that allows accumulation in tumor tissue; (2) recognition of glucose residues by GLUT receptors that are overexpressed at the tumor level; and (3) overexpression of reductase enzymes, characteristic of hypoxic microenvironments, that bioactivate FNZ.

## 5. Conclusions

The prepared pristine and glycosylated polymeric micelles formulations were inexpensive, easily prepared and scalable with potential application for the solubilization and stabilization of drugs with poorly soluble nature, such as FNZ, also preventing the drug attrition in early stages of research.

On the other hand, it is noteworthy that all pristine copolymers that formed PMs are approved by the FDA, which would facilitate the steps for a future clinical application [62]. Our goal at this stage was to demonstrate the potential of anti-tumoral activity of nanoencasulated fluorophenazine in vivo within pristine and glucosyltated PMs. Due to the aforementioned, we proposed these nanosystems loaded with FNZ as potential agents for therapeutic purpose breast cancer.

## Figures and Tables

**Figure 1 polymers-14-00071-f001:**
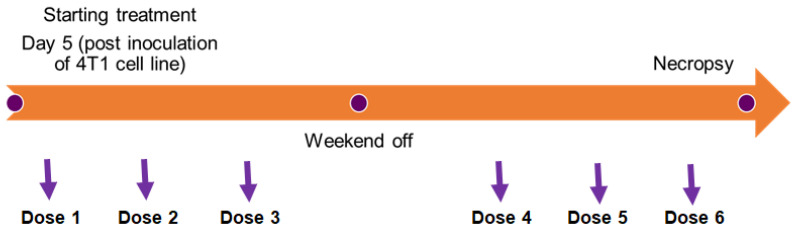
Nanoformulations administration schedule.

**Figure 2 polymers-14-00071-f002:**
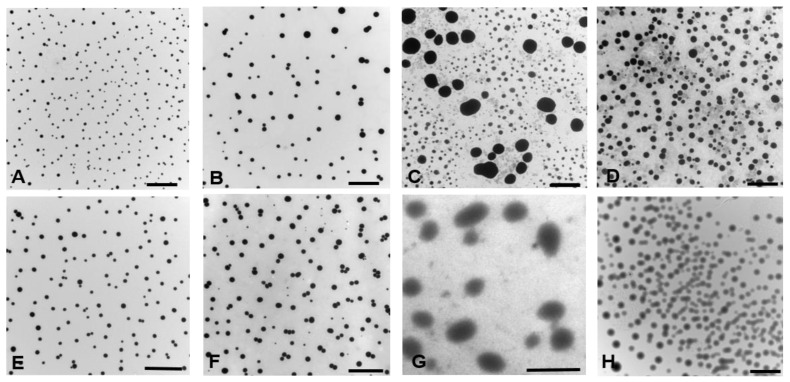
TEM micrographs of T1307:F127 (**A**), T1307:F127/FNZ (**B**), F127:LPS (**C**), F127:LPS/FNZ (**D**), F127-Glu (**E**), F127-Glu/FNZ (**F**), T1307-Glu (**G**) and T1307-Glu/FNZ (**H**). Note: Magnification: 6000×, except (**G**) 185,000×. Scale bars: 2 µm. Inset: average size distribution and Gaussian fitting of nanoparticles.

**Figure 3 polymers-14-00071-f003:**
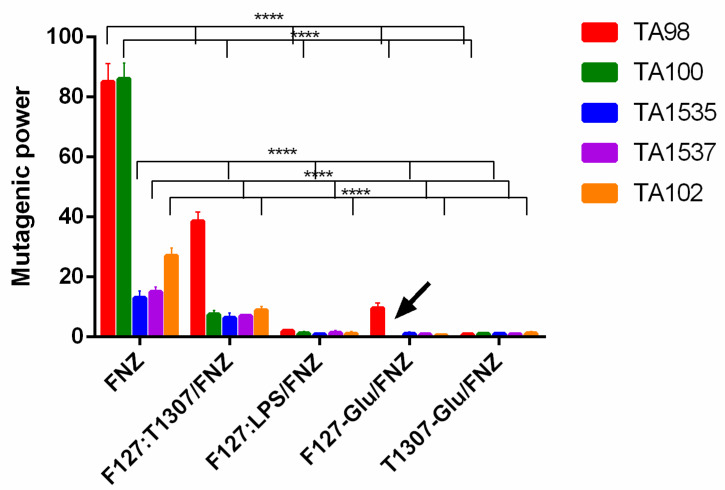
Mutagenic power as a function of the *Salmonella* typhimurium strains TA98, TA100, TA1535, TA1537 and TA102 exposed to FNZ or the different nanostructured formulations. Data are expressed as mean ± standard deviation (*n* = 3). Statically was analyzed by ANOVA “two way” followed by Tukey test. **** Statistically significant difference in mutagenic power (MP) between PMs/FNZ with respect to FNZ in each strain. Black arrow shows absence of mutagenicity.

**Figure 4 polymers-14-00071-f004:**
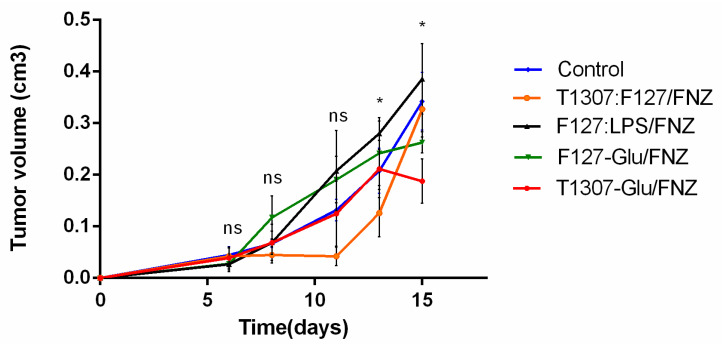
Evolution of tumor size during the different treatments. Data are expressed as mean ± standard deviation (*n* = 6). Statically was analyzed by ANOVA “two way” followed by Tukey test. * Statistically significant reduction in tumor volume of T1307:F127/FNZ treatment respect to control at day 14 and of F127-Glu/FNZ or T1307-Glu/FNZ treatments respect to control at day 16.

**Figure 5 polymers-14-00071-f005:**
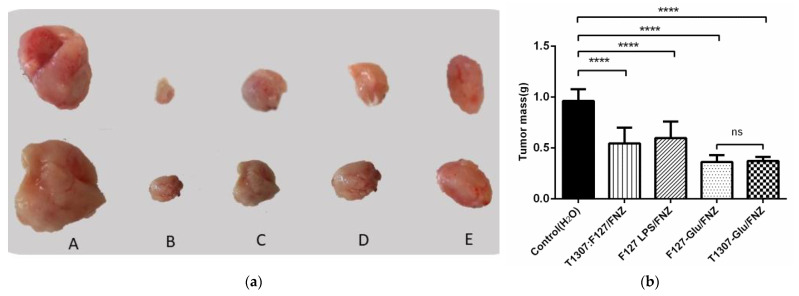
In vivo IP antitumoral evaluation with the different treatments: (**a**) Illustrative photographs of post treatment ex vivo tumors (showing two animals per treatment), with control (Milli-Q water, (A)), F127-Glu/FNZ (B), T1307-Glu/FNZ (C), T1307:F127/FNZ (D) and F127:LPS/FNZ (E) (20 mg/kg of FNZ). The tumor of (E) is shelled (non-solid); (**b**) Tumor masses at the end of IP treatments. **** Statistically very significant reduction of tumor weight comparing the control (tumor without treatment) with nanoformulations. ns there is no statistical significance. Note: animals per treatment = 6.

**Figure 6 polymers-14-00071-f006:**
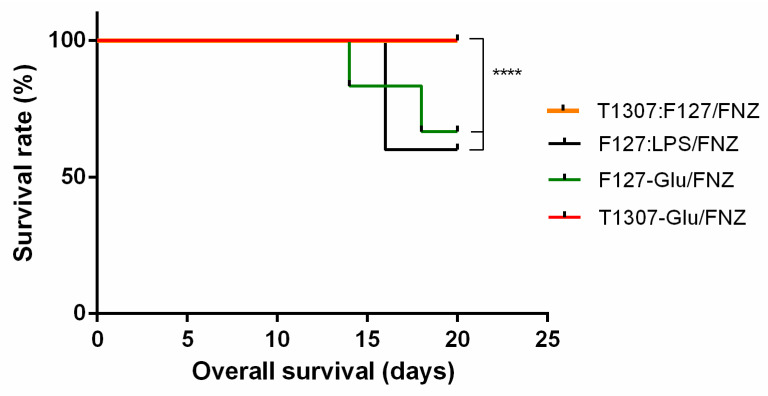
Animal survival in the different treatments (*n* = 6). The comparative analysis of the curves was carried out from the Kaplan-Meier analysis with the non-parametric log-rank test with a *p* < 0.05 (0.036), X^2^ > 3.84 (6.818) already described in experimental. **** Statistically very significant reduction in survival between F127:LPS/FNZ and F127-Glu/FNZ respect to T1307:F127/FNZ and T1307-Glu/FNZ.

**Table 1 polymers-14-00071-t001:** Characterization of PMs with and without FNZ cargo in aqueous medium.

Polymeric Nanosystems	S (μg/mL) ^1^	Fs ^2^	%EE ^3^	Peak 1 * D*_h_* (nm) ^4^	%Intensity	Peak 2 ** D*_h_* (nm)	%Intensity	PDI ^5^	Z-Pot (mV) ^6^
F127-Glu	-	-	-	410.5 ± 29.9	74.4 ± 1.6	40.5 ± 4.0	25.6 ± 1.6	0.359 ± 0.023	2.0 ± 0.2
F127-Glu/FNZ	1890 ± 25	9.1 ± 2.1	95 ± 2	700.0 ± 63.8	100.0 ± 0.0	-	-	0.335 ± 0.054	−1.6 ± 0.2
T1307-Glu	-	-	-	619.3 ± 63.9	57.0 ± 3.1	57.3 ± 8.7	43.0 ± 3.1	0.414 ± 0.140	−7.6 ± 0.7
T1307-Glu/FNZ	1494 ± 78	7.2 ± 1.8	74 ± 2	511.0 ± 54.4	100.0 ± 0.0	-	-	0.213 ± 0.077	−1.9 ± 0.5
T1307:F127	-	-	-	88.7 ± 2.5	76.7 ± 1.1	*** 4.5 (0.3)	11.3 ± 2.5	0.467 ± 0.018	−13.9 ± 0.6
T1307:F127/FNZ	2806 ± 142	13.5 ± 1.2	94 ± 1	59.7 ± 2.1	100.0 ± 0.0	-	-	0.180 ± 0.077	−2.8 ± 0.3
F127:LPS	-	-	-	546.6 ± 43.3	100.0 ± 0.0	-	-	0.317 ± 0.071	−15.5 ± 0.9
F127:LPS/FNZ	1733 ± 72	8.3 ± 1.6	87 ± 2	130.5 ± 6.30	100.0 ± 0.0	-	-	0.309 ± 0.011	−26.4 ± 0.5

^1^ S = Maximun aqueous solubility of FNZ in μg/mL in presence of PMs. ^2^ Fs (Solubility Factor) = S_PM_/S_water_, where S_PM_ and S_water_ are the apparent solubility of FNZ in the corresponding PMs and the experimental intrinsic solubility in water (pH = 5.8) (208 ± 23 μg/mL), respectively. ^3^ %EE = Percentage of encapsulation efficiency. ^4^ D*_h_* = Hydrodynamic diameter. ^5^ PDI = Polydispersity index. ^6^ Z-pot = Zeta potential. The determinations were done at 25 °C. * Major population %, ** Minor population %, *** Copolymer Unimers. Data are expressed as mean ± standard deviation (*n* = 3).

**Table 2 polymers-14-00071-t002:** Kinetic model fits for in vitro FNZ release from the nanoformulations.

Formulation	Order 0	Order 1	Higuchi	Korsmeyer-Peppas	
	*r^2^*	*r^2^*	*r^2^*	*r^2^*	*n*
F127-Glu/FNZ	0.9239 ± 0.0106	0.9916 ± 0.0008	0.9216 ± 0.0162	0.9743 ± 0.0087	1.47
T1307-Glu/FNZ	0.8220 ± 0.0098	0.9625 ± 0.0124	0.9927 ± 0.0108	0.9666 ± 0.0132	1.09
T1307:F127/FNZ	0.9485 ± 0.0003	0.9800 ± 0.0087	0.9865 ± 0.0020	0.9174 ± 0.0014	0.61
F127:LPS/FNZ	0.8684 ± 0.0049	0.8754 ± 0.0123	0.9707 ± 0.1243	0.9598 ± 0.0008	0.82

Data are expressed as mean ± standard deviation (*n* = 3).

**Table 3 polymers-14-00071-t003:** In vivo acute toxicity via IP.

Treatment	LD_50_ (mg/kg)	Irwin Test ^1^	Necropsies Observations ^2^
FNZ	1500	(+)	(-)
F127-Glu/FNZ	>2000	(+)	(-)
T1307-Glu/FNZ	>2000	(+)	(-)
T1307:F127/FNZ	>2000	(+)	(-)
F127:LPS/FNZ	>2000	(+)	(-)

^1^ The animals were analyzed at 0.5, 1.5, 3, 5, 12, 24, 48 and 72 h post-ip administration. When all the analyzed parameters were similar that control (saline treatment) it is labeled as (+), otherwise it is labeled as (-). ^2^ (-) Denotes the absence of organs alterations or organs adherences.

**Table 4 polymers-14-00071-t004:** Summary of the most relevant biochemical and hematological data of animals after in vivo treatments by serum of mice ^1^.

Treatment	Glycemia (g/L)	Urea (g/L)	Creatinine (mg/dL)	GPT (ALT) (UI/L)	GOT (AST) (UI/L)
F127-Glu/FNZ	0.74 ± 0.13	0.34 ± 0.1	0.16 ± 0.06	145 ± 12.3	364 ± 28.1
T1307-Glu/FNZ	0.82 ± 0.3	0.82 ± 0.3	0.22 ± 0.7	18 ± 2.4	122 ± 23.4
T1307:F127/FNZ	0.79 ± 0.18	0.32 ± 0.1	0.28 ± 0.05	39 ± 7.2	102 ± 14.3
Untreated	0.49± 0.12	0.48 ± 0.15	0.12 ± 0.02	147 ± 22.3	637 ± 88.1
Healthy animal	1.80 ± 0.09	0.34 ± 0.09	0.22 ± 0.11	36 ± 7.8	115 ± 8.02
	**hemoglobin (g/L)**	**MCH ^2^ (pg/cell)**	**platelets (cell/µL)**	**eosinophils (cell/µL)**	**basophils (cell/µL)**
F127-Glu/FNZ	12.9 ± 3.2	19.3 ± 1.9	876 ± 38.4	0.0 ± 0.0	0.3 ± 0.1
T1307-Glu/FNZ	12.9 ± 2.1	13.9 ± 1.0	768 ± 64.0	0.1 ± 0.0	0.0 ± 0.0
T1307:F127/FNZ	12.9 ± 0.9	15.4 ± 1.4	1259 ± 145.1	0.0 ± 0.0	0.0 ± 0.0
Untreated	13.9 ± 0.9	92.3 ± 13.8	1000 ± 233.3	0.38 ± 0.02	0.0 ± 0.0
Healthy animal	13.1 ± 1.3	16.8 ± 5.4	527 ± 87.0	0.6 ± 0.14	0.0 ± 0.0

^1^ The rest of the analyzed parameters (not shown) are in the normal ranges. ^2^ MCH: mean corpuscular hemoglobin. Data are expressed as mean ± standard deviation (*n* = 6).

## Data Availability

Not applicable.

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
