# Peer review of "Development and Evaluation of 2-Amino-7-Fluorophenazine 5,10-Dioxide Polymeric Micelles as Antitumoral Agents for 4T1 Breast Cancer"

_polymers, 2021, doi:10.3390/polym14010071_

Round 1
Reviewer 1 Report
Paper titled (Development and Evaluation of 2-Amino-7-Fluorophenazine 2 5,10-Dioxide Polymeric Micelles as Therapeutic Agents). By Lecot et al studied the synthesis of novel polymers for treating breast cancer. I have the following recommendations: 1- Title: please indicate which disease you are providing therapeutic options to? 2- Tables : write SD as +/- not between practice. 3- Introduction: reduce the first paragraph to half and add something about the cell line you used. 4- In table 4: I do not agree with the authors to use shades for values that are near or not near. In scientific research, we rely on statistics and differences were dtermined ONLY by statistical tests. Please revise the table and others in this regard. 5- Discussion is very short, some aspects like the toxicity of the naoparticles compared to the parent drug was not discussed. 6- How authors interpret the survival curve? please start from 0 % at x axis 7- Data in figure 3 are not stat analyzed 8- Methods: determine in stat anlysis which parameters analyzed bu one-way ANOVA and which by 2-way ANOVA? & what was teh rational for this 9- In figure legends or Table footnotes: mention the type of data (mean, median,...etc) and which stat test was applied Also add significance symbols and identify themAuthor Response
Please see the attachment

Reviewer 2 Report
This article have studied different formulations for the delivery of FNZ as a anticancer drug, the formulations have been thoroughly characterized, the in-vitro and in-vivo data look very promising. I would recommend publication after addressing the following comments.
- Considering recent advances in polymer based nano-carriers, the authors should have cited key references in the past five years instead of the old ones, ref 28-32 are too old and not representative of hydrophobic drug delivery using amphiphilic polymers. references such as 1. Angewandte Chemie - International Edition, 2020, 59(26), 10456–10460. https://doi.org/10.1002/anie.202002748, 2. Chemical Reviews 2018. 118, 14, 6844–6892. American Chemical Society. https://doi.org/10.1021/acs.chemrev.8b00199. 3. Pottanam Chali, S., & Ravoo, B. J. (2020). (Vol. 59, Issue 8, pp. 2962–2972). Wiley-VCH Verlag. https://doi.org/10.1002/anie.201907484. 4. Gao, J., Dutta, K., Zhuang, J., & Thayumanavan, S. (2020). Angewandte Chemie International Edition, 59(52), 23466–23470. https://doi.org/10.1002/anie.202008272 5. Liu, Y., Yang, G., Baby, T., Tengjisi, Chen, D., Weitz, D. A., & Zhao, C. X. (2020). Angewandte Chemie - International Edition, 59(12), 4720–4728. https://doi.org/10.1002/anie.201913539
- Magnification of TEM images should be provided in the methods section
- For the in-vivo anti-tumor evaluation, if how many mice were tested per treatment group? Replicates of tumor photographs should be provided in figure 5a
Reviewer 3 Report
The manuscript titled “Development and evaluation of 2-amino-7-fluorophenazine 5,10-dioxide polymeric micelles as antitumoral agents for 4T1 Breast Cancer” by Lecot et al. has been reviewed where the authors have demonstrated the potential of anti-tumoral activity of nanoencapsulated fluorophenazine. The work is well presented and supported with sufficient experimental investigations. However, the work seems to be weak regarding the characterization part. Hence, I believe that addressing the following comments will improve this aspect of the work.
- A general comment is that the manuscript needs improvement of the English language improvement as some of the sentences are not easily understandable.
- Line 123: The lineal calibration curve needs to be provided in a supplementary file.
- Line Nos. 148-155: The DLS data (size distribution, zeta potential and polydispersity index) needs to be provided as supplementary data. The data are provided in Table 1 but the raw data (plots off size distributions) obtained using DLS are not provided. Hence, the data could not be confirmed.
- In figure 2, can size distribution of particles for each case be provided as plot for each image? So that the homogeneity can be evaluated.
The above-mentioned points should be addressed before the work is considered for publication.
Author Response
Montevideo, 17 December 2021
Mr. Logan Sheng
Assistant Editor
Polymers Editorial Office Editor
Polymers
Dear Logan Sheng
I would like to thank you and the reviewers for the thorough and constructive evaluation of our manuscript initially entitled “Development and evaluation of 2-amino-7-fluorophenazine 5,10-dioxide polymeric micelles as therapeutic agents” by Lecot et. al. that I submitted for publication in the special issue entitle “State-of-the-Art Polymer Science and Technology in Uruguay (2021,2022)" that will be published in the journal Polymers.
In the new revised version (Round 2 for Reviewer 1, and Round 1 for Reviewer 3), we answered and incorporated the reviewer comments/recommendations.
Reviewer #1 (ROUND 2)
Unfoutrnately, authors did not reply to the previous comments, no statistical analysis is shown in some instances although mentioning they performed stat analysis and I find this paper should be rejected.
Response: This comment from reviewer 1 is very strange since ALL the previous recommendations were taken into account. They are repeated below, in case there was an error in the re-evaluation process by the reviewer 1 and in this new version (third version) we highlight in yellow the previous modifications incorporated according to reviewer 1 recommendations.
Paper titled (Development and Evaluation of 2-Amino-7-Fluorophenazine 2 5,10-Dioxide Polymeric Micelles as Therapeutic Agents). By Lecot et al studied the synthesis of novel polymers for treating breast cancer. I have the following recommendations:
1- Title: please indicate which disease you are providing therapeutic options to?
Response: We thank the reviewer very much for the positive evaluation of the manuscript and this suggestion. We incorporate “as antitumoral agents for 4T1 Breast Cancer” in the title.
2- Tables: write SD as +/- not between practice.
Response: Thanks for the recommendation to the reviewer. We have corrected it.
In this new version the +/- symbols were incorporated to clarify.
3- Introduction: reduce the first paragraph to half and add something about the cell line you used.
Response: Thanks for the recommendation to the reviewer. We have corrected it.
4- In table 4: I do not agree with the authors to use shades for values that are near or not near. In scientific research, we rely on statistics and differences were determined ONLY by statistical tests. Please revise the table and others in this regard.
Response: Thanks for the recommendation to the reviewer. We have corrected it.
5- Discussion is very short, some aspects like the toxicity of the nanoparticles compared to the parent drug was not discussed.
Response: Thanks for the recommendation to the reviewer. We have deepened the discussion.
6- How authors interpret the survival curve? please start from 0 % at x axis
Response: Thanks for the recommendation to the reviewer. We have corrected it.
7- Data in figure 3 are not stat analyzed
Response: This was a mistake in the original version. We have incorporated the analysis of the Figure 3.
8- Methods: determine in stat anlysis which parameters analyzed bu one-way ANOVA and which by 2-way ANOVA? & what was teh rational for this
Response: This was a mistake in the original version. We clarified that only two-way ANOVA was used for all the analyses.
9- In figure legends or Table footnotes: mention the type of data (mean, median, etc) and which stat test was applied Also add significance symbols and identify them.
Response: Thanks for the recommendation to the reviewer. We have corrected it.
Reviewer #3 (ROUND 1)
- A general comment is that the manuscript needs improvement of the English language improvement as some of the sentences are not easily understandable.
Response: It was done in the previous revised version.
- Line 123: The lineal calibration curve needs to be provided in a supplementary file.
Response: It was included in the supplementary file.
- Line Nos. 148-155: The DLS data (size distribution, zeta potential and polydispersity index) needs to be provided as supplementary data. The data are provided in Table 1 but the raw data (plots off size distributions) obtained using DLS are not provided. Hence, the data could not be confirmed.
Response: It was included in the supplementary file.
- In figure 2, can size distribution of particles for each case be provided as plot for each image? So that the homogeneity can be evaluated.
Response: It was included in figure 2.
I hope you will find the revised version of the manuscript suitable for publication. I am looking forward hearing from you.
Sincerely yours,
Dr. Nicole Lecot

Round 2
Reviewer 1 Report
Unfoutrnately, authors did not reply to the previous comments, no statistical analysis is shown in some instances although mentioning they performed stat analysis and I find this paper should be rejected.
Author Response

(The authors gave the same response as above.)
